# A Framework for Reviewing Silvopastoralism: A New Zealand Hill Country Case Study

**Thomas H. Mackay-Smith** [1,*] **, Lucy Burkitt** [1] **, Janet Reid** [1] **, Ignacio F. López** [1] **and Chris Phillips** [2]

1    School of Agriculture and Environment, Farmed Landscapes Research Centre, Massey University, Palmerston North 4442, New Zealand; l.burkitt@massey.ac.nz (L.B.); j.i.reid@massey.ac.nz (J.R.); i.f.lopez@massey.ac.nz (I.F.L.)
2    Manaaki Whenua-Landcare Research, P.O. Box 69040, Lincoln 7640, New Zealand; phillipsc@landcareresearch.co.nz
*    Correspondence: t.mackaysmith@massey.ac.nz

**Abstract:** Silvopastoral systems can be innovative solutions to agricultural environmental degradation, especially in hilly and mountainous regions. A framework that expresses the holistic nature of silvopastoral systems is required so research directions can be unbiased and informed. This paper presents a novel framework that relates the full range of known silvopastoral outcomes to bio-physical tree attributes, and uses it to generate research priorities for a New Zealand hill country case study. Current research is reviewed and compared for poplar (*Populus* spp.), the most commonly planted silvopastoral tree in New Zealand hill country, and kānuka (*Kunzea* spp.), a novel and potentially promising native alternative. The framework highlights the many potential benefits of kānuka, many of which are underappreciated hill country silvopastoral outcomes, and draws attention to the specific outcome research gaps for poplar, despite their widespread use. The framework provides a formalised tool for reviewing and generating research priorities for silvopastoral trees, and provides a clear example of how it can be used to inform research directions in silvopastoral systems, globally.

**Keywords:** New Zealand; hill country; poplar; kānuka; agroforestry; silvopasture; soil conservation; erosion; ecosystem services



## 1. Introduction

Agroforestry is a land use where at least two plant species interact biologically, at least one of the plant species is a woody perennial (typically trees), and at least one of the plant species is managed for forage, annual or perennial crop production [1]. Silvopastoral systems are a type of agroforestry system where trees are integrated into a pastoral system. Silvopastoral systems are commonly adopted in environmentally sensitive areas to mitigate landscape degradation [2–4]. Many of these are hilly or mountainous, and include for example the poplar (*Populus* spp.) and willow (*Salix* spp.) silvopastoral systems of New Zealand hill country [2], the ñire (*Nothofagus antarctica* (G. Forst.) Oerst.) system of Patagonia [3], and the oak (*Quercus* spp.) silvopastures of California [5], the Indian Himalayas [6], and Spain (the *dehesa* system) [7].

Silvopastoral systems are inherently complex and result in many ecological, economic, and cultural outcomes within the agricultural system. To fully understand and appreciate silvopastoral systems, it is important that research spans as many of these outcomes as possible. If only specific outcomes are studied, research or tree planting choices may be biased towards these narrowly selected outcomes, and other potential benefits may be overlooked or underappreciated. Moreover, if the maximum benefits of silvopastoral tree plantings are to be realised and plantings are to be justified, their full range of known benefits and costs must be compared.

As an example of this, in the hill country of New Zealand (an area characterised by hilly or steep land (> 15°), being below 1000 m asl, and pastoral farming as its main land

use) [8], there is a narrow research focus on the principal silvopastoral tree genera that are planted (poplar and willow). The focus is on pasture production, soil conservation, and establishment ease [2,9,10], with soil conservation value and establishment ease primarily informing planting decisions.

These genera have been shown to be highly effective as soil conservation trees [11–13], and can be planted easily as 2–3 m unrooted coppiced poles with sheep and small cattle grazed immediately after establishment [2,14]. Nevertheless, in hill country, as far as we are aware, there has been no research on other silvopastoral outcomes such as bio-diversity conservation value, wind run reductions, shelter value comparison between species or genera, and impacts on catchment discharge rates (in typical planting densities of 20–200 tree ha$^{-1}$), among others that will be highlighted in this paper. Historically, other genera have been overlooked because only a few factors have been considered in planting decisions, even though many alternative species may be more suitable in certain situations, or their overall benefits greater than those of poplar or willow.

Wood [15] presented a framework, which itself was adapted from Von Carlowitz [16], that provides a useful way of looking at the range of outcomes within agroforestry systems. The authors split trees into their bio-physical attributes and related these to 'performance' in an agroforestry system (Table 1). Dividing trees into their bio-physical attributes is useful because it helps show why a tree may be contributing to a positive or negative silvopastoral outcome. This means that alternative trees can be selected based upon their attributes, and silvopastoral species or genera research or tree planting choice can then be optimised for specific silvopastoral systems, based upon a system's outcome needs.

However, the framework presented by Wood [15] is focused on agroforestry rather than silvopastoral systems, and takes a narrow tree performance view on silvopastoral system outcomes, therefore missing their holistic nature. Because of this primary reason, and others that will be explored in the next section, this paper presents a new framework that identifies and links bio-physical attributes to system outcomes like Wood [15], but does so for a silvopastoral system. It also expands the outcomes to account holistically for the full range of known silvopastoral outcomes. Section 2 will present this framework and explain in more detail how it differs from Wood [15], and why these changes are necessary.

We believe that our new framework will appeal to multiple groups. Firstly, it provides a standardised methodology for the research community to review silvopastoral research, and to identify research priorities that will improve the understanding of specific silvopastoral systems. Additionally, it will enable researchers to review trees in relation to all their known potential outcomes, and reduce research biases on specific outcomes, as has been the case in New Zealand hill country to date. The second half of this paper will illustrate the framework being used in this way, and will compile current knowledge for poplar, the most commonly planted and researched silvopastoral tree genus in New Zealand, and kānuka (*Kunzea* spp.), a genus that has received little attention in a hill country silvopastoral context. Based on the framework, the genera will be assessed, reviewed and compared across their full range of known benefits and costs.

Secondly, the framework will provide an opportunity for practitioners and land managers to see the full range of known interactions within a silvopastoral system. It will also clearly highlight the holistic nature of silvopastoral systems, and reduce the focus on only specific outcomes, as has been the case in New Zealand hill country. When trees have been reviewed and compared, this comparison can then be used by land managers to decide which tree may be best for their specific situation, depending on their requirements. Finally, in time, a unit of value could be added to the different outcomes in the framework. This would allow researchers, land managers, and land owners to quantifiably discriminate which tree may be best for a specific situation. This however, is beyond the scope of this paper.

Silvopastoral systems are complex, comprising multiple inter-related components. A framework that captures this complexity is fundamental to ensure that the full potential of silvopastoral trees are researched, realised, and appreciated. The framework will be a

valuable tool for those selecting and researching silvopastoral trees, especially in hilly or mountainous regions.

**Table 1.** An agroforestry framework relating tree attributes to 'performance' in an agroforestry system reproduced from Wood [15], which was adapted from Von Carlowitz [16]. Copyright © 1990 John Wiley & Sons, Inc.

| Tree Attributes | Relationship to Performance in Agroforestry Systems |
|---|---|
| Height | Ease of harvesting leaf, fruit, seed, and branchwood; shading or wind effects |
| Stem form | Suitability for timber, posts, and poles; shading effects |
| Crown size, shape, and density | Quantity of leaf, mulch, and fruit production; shading or wind effects |
| Multistemmed habit | Fuelwood and pole production; shading or wind effects |
| Rooting pattern (deep or shallow, spreading or geotrophic) | Competitiveness with other components, particularly resource sharing with crops; suitability for soil conservation |
| Physical and chemical composition of leaves and pods | Fodder and mulch quality; soil nutritional aspects |
| Thorniness | Suitability for barriers or alley planting |
| Wood quality | Acceptability for fuel and various wood products |
| Phenology (leaf flush, flowering, and fruiting) and cycle (seasonality) | Timing and labor demand for fruit, fodder, and seed harvest; season of fodder availability; barrier function and windbreak effects |
| Di = or monoeciousness | Sexual composition of individual species in community (important for seed production and pollen flow) |
| Pest and disease resistance | Important regardless of function |
| Vigor | Biomass productivity, early establishment |
| Site adaptability and ecological range | Suitability for extreme sites or reclamation uses |
| Phenotypic or ecomorphological variability | Potential for genetic improvement, need for culling unwanted phenotypes |
| Response to pruning and cutting management practices | Use in alley farming, or for lopping or coppicing |
| Possibility of nitrogen fixation | Use in alley farming, planted fallows, or rotational systems |

## 2. A Framework for Assessing Silvopastoralism

Figure 1 shows the new framework for silvopastoralism, which outlines all the known interactions within a silvopastoral system between a tree's bio-physical attributes and system outcomes. The following section explains in more detail how this new framework improves on the original framework by Wood [15].

As Wood [15]'s framework was designed for agroforestry systems in general and not silvopastoral systems, it places little emphasis on the interactions fundamental to a silvopastoral system. These include interactions between trees and livestock, and between the grazing livestock, soil, and pasture.

Many environmental, management, and cultural outcomes associated with silvopastoral systems are also lacking in the original framework. In the new framework, outcomes are expanded to include the following environmental outcomes: 'water and nutrient gains or losses', 'biodiversity interactions (excluding livestock and the forage crop)', 'greenhouse gas implications', and 'longevity of the tree'; management outcomes: 'costs and ease of establishment', 'special qualities reducing animal interactions with the tree', 'longevity of the tree', and 'ability to refine the tree form for improved silvopastoral outcomes'; and cultural outcomes: 'livestock shelter' ('livestock shelter' is a cultural outcome in terms of animal health reasons and a production outcome in terms of live weight increase reasons), 'cultural values', and 'aesthetics'.

In addition to outcomes, the new framework also includes additional attributes of specific relevance to hill country silvopastoral systems, including 'growth rate', 'establishment method (seedling, cutting, pole)', and 'water use'. In hill country, silvopastoral systems commonly need to be established, as trees are generally lacking, so a tree's growth rate and establishment form is a key consideration in planting decisions. Moreover, the interaction between the tree and pasture in terms of water is also important, an attribute lacking in Wood [15]'s framework.

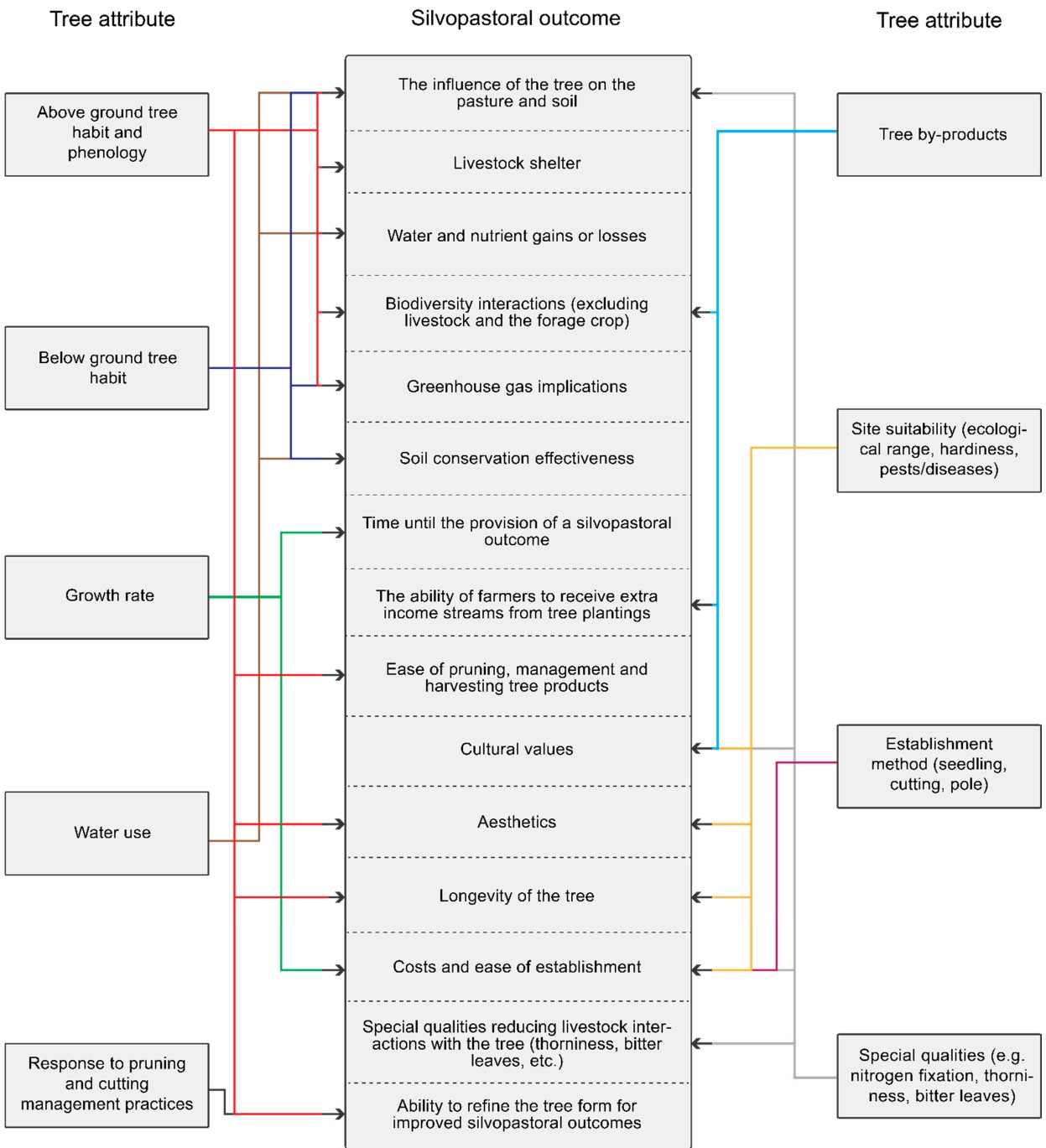

**Figure 1.** A framework of the interactions within a silvopastoral system between tree attributes and silvopastoral outcomes.

To improve the simplicity of Wood [15]'s framework, multiple attributes have been grouped into one attribute in the new framework. For example, the attributes 'height', 'stem form', 'crown size, shape and density', 'multistemmed habit', 'phenology (leaf flush, flowering and fruiting) and cycle (seasonality)', and 'Di = or monoeciousness' are encompassed into the broader attribute: 'above ground tree habit and phenology'. Moreover, 'pest and disease resistance', 'vigor', 'site adaptability and ecological range', and 'phenotypic or ecomorphological variability' are grouped to form one attribute: 'site suitability (ecological range, hardiness, pests/diseases)'.

A 'special qualities' attribute was also added to the framework so any unique tree qualities in other silvopastoral systems can be incorporated into the framework.

Wood [15]'s framework was primarily developed to inform the selection of trees for agroforestry systems. The new framework can also be used in this way, but we extend the use of our framework and use it to comprehensively collate research and knowledge on particular trees. In doing so, this paper clearly highlights the practical use of the framework to researchers for assessing and comparing silvopastoral trees.

## 3. Using the Framework: A New Zealand Hill Country Case Study

The following section will illustrate how the framework can be used to review two silvopastoral systems and generate research priorities in a degraded, but economically important, hilly and mountainous region.

As definitions of New Zealand hill country vary [8], so do area estimations, but one estimate of the pastoral farming land area in hill country is 5.2 million ha (19.4% of New Zealand's land mass) [17]. Much of this hill country is marginal agricultural land, associated with reduced organic matter, nutrient levels and water holding capacities, resulting in many areas having a low productive potential [18]. Due to the highly topographic and treeless nature of hill country, soil erosion and surface runoff discharge rates are high [8,19].

These poor conditions have multiple ramifications for New Zealand. High sediment loads alter local floral and faunal streambed habitats [8], reduce river clarity, and reduce the soil base of hill country farms. Nitrogen (N) and phosphorus (P) losses with sediment encourage algal growth [20,21], further degrading river habitats and the quality of water supplies [22]. Furthermore, elevated surface water discharge results in elevated flood severity and risk [23].

### 3.1. Poplar and Willow

The principal soil conservation intervention in New Zealand is tree planting, specifically aimed at the mitigation of shallow mass movement events (shallow soil slips), earthflows and gully erosion [12,24]. Space-planted poplar and willow are the main genera grown, planted at densities that generally range from 20 to 200 trees ha$^{-1}$ [2,9] (Figure 2). Afforestation is additionally used for soil conservation, in the form of exotic forestry plantations (principally radiata pine (*Pinus radiata* D. Don) at densities ~1200 trees ha$^{-1}$) [2], native mānuka (*Leptospermum scoparium* J.R. Forst and G. Forst) plantations for honey production [2], or native forest via unmanaged regeneration or native seedling establishment.

Poplar and willow have been extensively researched, including reviews by Benavides et al. [9], Kemp et al. [2], and Basher et al. [4]. They have been shown to be highly effective as soil conservation trees [11–13], and can be planted easily as 2–3 m unrooted coppiced poles with sheep and small cattle grazed immediately after establishment [2,14]. Nevertheless, a 40-year tree life is recommended as branch breaking is common [25], reducing the long-term soil conservation or carbon sequestration potential of each tree compared to if the tree was not felled. Additionally, the negative effects of poplars on pasture growth are well established [2,9] and there is little evidence they improve soil properties beneath their canopies [26,27].

In terms of other species, Devkota et al. [28] studied the canopy effect of Italian gray alder (*Alnus cordata* (Loisel.) Duby) on soil and pasture, and Australian blackwood (*Acacia melanoxylon* R.Br.) and Eucalyptus (*Eucalyptus* spp.) have also been studied in the context of timber production, pasture production, soil properties and landslide mitigation [11,29–31]. Many trees and shrubs have been researched for their potential use as fodder trees including research on poplars and willows (e.g. [32–34]), as well as tagasaste (*Cytisus proliferus* L.f.) and saltbush (*Atriplex halimus* L.) [35,36], among others [35]. Nonetheless, poplar and willow remain the dominant silvopastoral system in hill country, despite their constraints.

### 3.2. Kānuka

Kānuka is a native and successional genus that grows readily in New Zealand hill country [13], and has many attributes that mean it has the potential to perform well as a silvopastoral tree. Kānuka has been split into 10 endemic New Zealand species [37], although Heenan et al. [38] provides evidence that questions this 10-species description. Nevertheless,

as of 2021, 10 are still recognised. These 10 species occupy different ecological niches and geographical extents [13]. Seven of these species (*K. amathicola* de Lange et Toelken, *K. ericoides* (A.Rich.) Joy Thomps., *K. linearis* (Kirk) de Lange & Toelken, *K. robusta* de Lange et Toelken, *K. salterae* de Lange, *K. serotina* de Lange et Toelken, *K. triregensis* de Lange) are trees that can reach greater than 10 m in height [37] and are the most suitable for use in a silvopastoral system. Most people collectively refer to these species by their common name, kānuka. This paper will use the term kānuka and is specifically referring to the seven kānuka species that are greater than 10 m in height when growing in native forest.

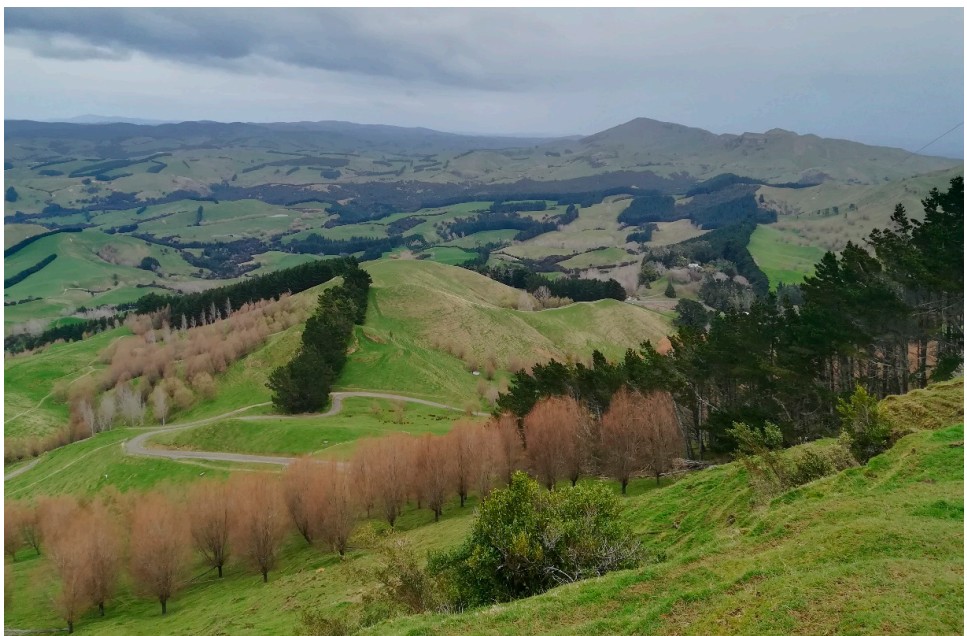

**Figure 2.** A typical North Island New Zealand hill country landscape 25 km north east of Dannevirke, in the Manawatū-Whanganui region. Willows can be seen space-planted in pastures at the bottom of the slope directly beneath the photographer. The photograph was taken by the lead author.

In most places, kānuka, along with mānuka and gorse (*Ulex europeaus* L.), are the first woody perennial species to colonise unmanaged pasture in New Zealand hill country [39,40]. When kānuka grows on this unmanaged pasture, the predominant practice is to clear the kānuka to produce treeless pastures [41]. However, this paper demonstrates that kānuka has many beneficial attributes in a silvopastoral system and that thinning instead of clearing higher density kānuka stands [13], or even space-planting the tree on hill country pastures, should be encouraged.

### 3.3. Reviewing Current Knowledge for Poplar and Kānuka

Drawing on existing research and knowledge, poplar, the most commonly planted silvopastoral tree in hill country, and kānuka are now reviewed and compared according to the framework tree attributes (Table 2) and system outcomes (Table 3). Information compiled in Tables 2 and 3 comes from different sources. The first are visible factors, such as the heights of unmanaged trees growing in hill country. This is information in Table 2 that has no references. The second are projected interactions that have been logically inferred based upon known tree attributes (e.g., an evergreen tree will provide more shelter in winter than a deciduous tree). These are sections in Table 3 under the label 'likely outcome' where there is 'no evidence' for a specific outcome. The final source is literature, either peer-reviewed scientific research or reports. These are labelled as 'evidence for' and 'evidence against' in Table 3. Table 2 does not contain 'evidence for' or 'evidence against' descriptions because tree attributes themselves are not positive or negative, but it is the resultant outcomes that are positive or negative to the user of the silvopastoral system.

**Table 2.** Tree attributes for poplar (*Populus* spp.) and kānuka (*Kunzea* spp.) in a New Zealand hill country silvopastoral system. Tree attributes have been adapted from Wood [15]. The photographs were taken by the lead author.

| Tree Attribute | Poplar (*Populus* spp.) Attribute | Priority Research Area | Kānuka (*Kunzea* spp.) Attribute | Priority Research Area |
|---|---|---|---|---|
| Above ground tree habit and phenology | Current cultivars planted in the 1960s and 1970s are >30 m in height.<br>Crowns are large and uncompact. Older cultivars often have large branches extended; some are multistemmed. Newer cultivars have been developed which grow as a single, straighter stem.<br>Deciduous.<br> | Yes—an understanding of the form of newer poplar cultivars when they are fully-grown would be informative. | When growing isolated in hill country, kānuka are 8–20 m in height.<br>Compact crowns. Stems can be multi- or single-stemmed. Many branches when unmanaged. The form of kānuka varies with tree density, growing taller and thinning in higher densities.<br>Evergreen.<br> | No |

**Table 2.** *Cont.*

| Tree Attribute | Poplar (*Populus* spp.) Attribute | | Priority Research Area | Kānuka (*Kunzea* spp.) Attribute | | Priority Research Area |
|---|---|---|---|---|---|---|
| Below ground tree habit | For three 11.5-year-old poplar trees on a 17° hill country site at densities of 156 tree ha$^{-1}$, maximal lateral root extension ranged from 8.0–12.0 m. Mean tensile strength of 44.0 (minimum: 11.1 MPa; maximum: 114.3 MPa). The total root length of a 9.5-year-old poplar tree was found to be 663.5 m with a root biomass of 17.9 kg. The lateral root extension, root biomass and total root length of 'fully-grown' poplar trees on hill country > 25.0° would be valuable. | McIvor et al. [42,43]; Watson et al. [44] | No | Only kānuka growing in high density forest stands (~3000–16,000 stems ha$^{-1}$) have been studied. Fifteen 16-year-old trees growing at 12,800 stems ha$^{-1}$ had a maximum root length of 4.5 m. Fifteen 32-year-old trees growing at 3900 stems ha$^{-1}$ had a maximum root length of 6.1 m Mean tensile strength of 34.1 MPa (minimum: 18.2 MPa; maximum: 75.8 MPa). In another high-density stand (3000 stems ha$^{-1}$), the total root length of one fully-grown kānuka tree 9.5 m in height was shown to be 123.2 m, have a root biomass without the stump of 11. 8 kg and a lateral root spread of 2.8 m. | Watson et al. [45,46]; Watson and Marden [47]; Watson and O'Loughlin [48] | Yes—research on the root distribution of kānuka growing at typical hill country silvopastoral densities (20–200 tree ha$^{-1}$) is required. |

**Table 2.** *Cont.*

| Tree Attribute | Poplar (*Populus* spp.) Attribute | | Priority Research Area | Kānuka (*Kunzea* spp.) Attribute | | Priority Research Area |
|---|---|---|---|---|---|---|
| Growth rate | On a 21–35° slope, the mean height of 268 poplar poles was just under 3.0 m after 12 months, ~3.5 m after 24 months and ~5.3 m after 45.0 months. Start heights were not given by the authors so yearly growth rates could not be calculated. 5, 7 and 9.5 years old trees had heights of 7.0, 9.5 and 13.3 m, respectively, on a 17° hill country site. This equates to a ~1.3 m year$^{-1}$ growth rate (accounting for the 1.4 m start height of the poles). | Marden and Phillips [49]; McIvor et al. [42] | No | Initial growth rates are often 0.7–0.8 m year$^{-1}$ in sheltered and high fertility sites, and 0.4–0.5 m year$^{-1}$ in poorer sites. This data was collected from interviews, and was not stated to be quantitatively studied in the report. | Boffa Miskell Limited [50] | Yes—quantitative information on growth rates in contrasting conditions, as well as at 20–200 tree ha$^{-1}$ densities, is required. |
| Water use | Four trees were shown to have an average water use of 180.1 L day$^{-1}$ during spring, which equated to 1.2 mm day$^{-1}$. One of these trees had a water use of 417.0 L day$^{-1}$. | Guevara-Escobar et al. [51] | No | | | Yes—the water use of kānuka is unknown. |
| Response to pruning and cutting management practices | Responds well to pruning when the trees are young, as well as coppicing and pollarding. | Charlton et al. [25] | No | | | Yes—the response to management is unknown. |

| Tree Attribute | Poplar (*Populus* spp.) Attribute | Priority Research Area | Kānuka (*Kunzea* spp.) Attribute | Priority Research Area |
|---|---|---|---|---|
| Tree by-products | Wood—poplars can be pruned and harvested for timber. Fodder—leaves are excellent fodder for animals. Emissions trading scheme (ETS)—there is the potential for farmers to receive carbon credits (1 NZU = 1 tonne of sequestered $CO_2$) if the tree crown canopy is >30% in each hectare. | Charlton et al. [25]; Kemp at al. [2]; MPI [52] | Yes—research required to understand the density required to achieve a 30.0% canopy cover with poplar. | Wood—reported to be good firewood. Fodder—kānuka leaves are 0.5–2.5 cm, the tree doesn't have summer leaf flush as they are evergreen and the leaves are potentially bitter, so we tentatively suggest that the trees would be poor fodder quality. Honey—shown to have anit-bacterial, anti-viral, immunostimulatory and anti-inflammatory properties. Essential oil—kānuka essential oil has been shown to be an effective eco-friendly pesticide. Emissions trading scheme—potential exists for farmers to receive carbon credits (1 NZU = 1 tonne of sequestered $CO_2$) if the tree crown canopy is > 30% in each hectare. | Boffa Miskell Limited [50]; Bloor [53]; Gannabathula et al. [54]; Lu [55]; Tomblin et al. [56]; Kassimi et al. [57]; Park [58]; MPI [52] | Yes—research required to understand the density required to achieve a 30.0% canopy cover with kānuka. |

**Table 2.** *Cont.*

| Tree Attribute | Poplar (*Populus* spp.) Attribute | Priority Research Area | Kānuka (*Kunzea* spp.) Attribute | Priority Research Area |
|---|---|---|---|---|
| Site suitability (ecological range, hardiness, pests/diseases) | Exotic to hill country, although poplar in certain conditions can have a high survival rate when established in hill country.<br>For 300 hill country poplar poles deaths after 45 months, site factors (site conditions, socketing etc.) contributed to 28% deaths, and animal damage contributed to 12% of deaths.<br>After 6 years, survivability on six hill country farms ranged from 0% to 80% (slopes varied from 0% to 32% in the study). Although the reasons for death were not quantitively measured by the authors, reasons given include animal damage, poor planting, continued erosion, winter weather fronts and poor local site conditions. Fungus and rust can be issues, with more resistant clones the main mitigation strategy.<br>As branch breaking is common due to high winds in hill country, and fungus and rust can be issues, best management practice suggests felling and replanting the trees after 40 years. An understanding of the survival rate of poplars on different slope classes (especially the steepest hill country slopes) and in different environmental conditions would be informative, as well as more detailed quantitative information on the reasons for the low survival rates. | Marden and Phillips [49]; McIvor et al. [59]; Charlton et al. [25] | Yes—an understanding of the survival rates of poplar poles on the steepest, most erosion prone, hill country slopes would be helpful. | Native to hill country and already grows readily throughout hill country.<br>Kānuka is reported to potentially grow up to at least 160 years and possibly as old as 300–400 years.<br>Kānuka can grow in unfertile and moisture limited areas of hill country.<br>Kānuka are susceptible to myrtle rust as they are in the myrtle family, Myrtaceae.<br>Data on the survival percentages of kānuka in varying soil conditions is required, as well as how susceptible a kānuka silvopastoral system would be to myrtle rust. | Spiekermann et al. [13]; Boffa Miskell Limited [50] | Yes—quantitative data is lacking on the age to which kānuka grow at 20–200 tree ha$^{-1}$ densities in hill country, establishment survival rates and the system's susceptible to myrtle rust. |

**Table 2.** *Cont.*

| Tree Attribute | Poplar (*Populus* spp.) Attribute | Priority Research Area | Kānuka (*Kunzea* spp.) Attribute | Priority Research Area |
|---|---|---|---|---|
| Establishment method (seedling, cutting, pole) | Can be established as unrooted 1.0–3.0 m poles or stakes (0.5 m cuttings) which are sharpened and rammed into the ground. Sheep and small cattle can be grazed immediately. Large cattle can knock over and break poplar poles, so exclusion until the poles have established is recommended. Regular poplar poles that are planted in hill country normally take 2–3 years to produce, depending on the region, occupy a lot of land in their production and demand for them regularly outstrips supply. Understanding the establishment methods and survival rates of quicker to produce planting material (younger unrooted material or rooted material) that can be grown in a smaller amount of land with less water and lower costs would be helpful. | Marden and Phillips [49]; Phillips et al. [14]; Ian McIvor (personal communication, 26th October 2021) [60] | Yes—understanding the establishment of different planting material (younger unrooted material or rooted material) would be helpful. | With current planting technology and knowledge kānuka would need to be planted as seedlings and protected from animal browsing. Large cattle may require exclusion depending on the protection method. Protection with current technology would need to be strong 1.7 m plastic netting or a wire cage, supported by 2 Y posts for cattle, or by a Y post and a fibreglass rod for sheep. It is unknown at what age seedling protection can be removed. | Yes—little is known on the establishment of kānuka in hill country. |

**Table 2.** *Cont.*

| Tree Attribute | Poplar (*Populus* spp.) Attribute | Priority Research Area | Kānuka (*Kunzea* spp.) Attribute | Priority Research Area |
|---|---|---|---|---|
| Special qualities (e.g., nitrogen fixation, thorniness, bitter leaves) | No special qualities of note. | No | A key difficulty when establishing trees in hill country is livestock browsing or damaging the tree. Livestock exclusion from paddocks is often not possible. Some land managers state kānuka leaves are bitter, which may reduce or stop browsing by sheep and cattle during establishment. Evidence for this is kānuka is already found growing readily in many parts of unproductive hill country in the presence of animals. Fresh shoots or young seedlings from commercial nurseries are likely to be browsed. | Yes—more information on the relationship between kānuka leaves and livestock is required. |

**Table 3.** Silvopastoral outcomes for poplar (*Populus* spp.) and kānuka (*Kunzea* spp.) in a New Zealand hill country silvopastoral system. Tree outcomes have been adapted from Wood [15].

| Silvopastoral Outcome | Poplar (*Populus* spp.) Outcome | Priority Area for Research | Kānuka (*Kunzea* spp.) Outcome | Priority Area for Research |
|---|---|---|---|---|
| Influence of the tree on the pasture and soil | *Evidence against*<br>Pasture reduction beneath the canopy between 12% and 65% for poplar greater than 15 years old.<br>A relationship has been found between increased canopy closure and decreased pasture production.<br>Leaf smother has been shown to depress autumn grass growth beneath poplar canopies.<br>Poplars do not fix nitrogen.<br>One study found 33.0% less soil moisture beneath poplars when compared with open pasture in summer and autumn.<br>Another study found slightly more water in the top 15 cm in pasture away from poplar throughout the year, with the difference most pronounced in summer and autumn.<br>As pasture production and soil moisture has been shown to reduce under poplar, there is no evidence that wind-run reductions caused by poplar facilitate water conservation in the soil.<br>Found no evidence that poplar facilitate the build-up of organic matter, nitrogen, phosphorus or sulphate beneath their canopies between 0.0 and 7.5 cm at three sites with poplar trees > 28 years old.<br>Found varied results of soil organic matter, phosphorus and sulphate beneath fully developed poplar canopies between 0.0 and 15.0 cm compared to open pasture at two sites.<br>There is evidence that poplar increase exchangeable cations (calcium, potassium, magnesium, sodium) beneath their canopies, most likely because of the chemical composition of their leaves.<br>Along with light interception and autumn pasture smother, the water use of poplar could be contributing to the reduced pasture production beneath their canopies. | Reviewed by Benavides et al. [9]; Wall et al. [61]; Douglas et al. [62]; Kemp et al. [2]; Douglas et al. [63]; Guevara-Escobar et al. [64]; Guevara-Escobar et al. [26]; Wall [27]<br><br>No—there is a good understanding of how poplar influences the pasture and soil. | *No evidence*<br>*Likely outcome*<br>There has been no research on pasture production, and the constraints to pasture production, beneath kānuka in hill country. More research is required to produce any likely outcome predictions for the influence of kānuka on pasture, livestock and soil. Kānuka are evergreen, so this may have varying influences on the system when compared to poplar. Kānuka do not fix nitrogen. | Yes |

**Table 3.** *Cont.*

| Silvopastoral Outcome | Poplar (*Populus* spp.) Outcome | Priority Area for Research | Kānuka (*Kunzea* spp.) Outcome | Priority Area for Research |
|---|---|---|---|---|
| Livestock shelter | *No evidence*<br>*Likely outcome*<br>Trees will most likely provide less shelter to animals in winter than summer (poplars are deciduous).<br>The summer shelter will most likely be positive for animal grazing time in summer, and may reduce heat stress resulting in greater live weight growth of livestock.<br>The influence of poplar stem and branches on wind-run in winter may have positive influences in terms of reduced deaths and increased livestock live weight growth by reducing wind chill. | Yes | *No evidence*<br>*Likely outcome*<br>As kānuka are evergreen it is expected the trees will provide good shade and shelter to animals in summer and winter.<br>The summer and winter shelter will most likely be positive for animal grazing time throughout the year.<br>The influence of kānuka on wind-run in winter may have positive influences in terms of reduced livestock deaths and increased livestock live weight growth by reducing wind chill. | Yes |
| Water and nutrient gains or losses | *No evidence*<br>Hill country 20–200 tree ha$^{-1}$ densities have not been studied. | Yes | *No evidence*<br>It is unknown how kānuka impacts these system dynamics. | Yes |

Table 3. *Cont.*

| Silvopastoral Outcome | Poplar (*Populus* spp.) Outcome | | Priority Area for Research | Kānuka (*Kunzea* spp.) Outcome | | Priority Area for Research |
|---|---|---|---|---|---|---|
| Biodiversity interactions (excluding livestock and the forage crop) | *Evidence against*<br>Poplar were found to either reduce or maintain earthworm populations compared to equivalent open pasture positions. The three most abundant earthworms found beneath poplars were all exotic (*Aporrectodea caliginosa*, *A. longa*, *Lumbricus rubellus*).<br>*No evidence*<br>As far as we are aware, nothing is known on how poplar influence bird, insect and fungi populations.<br>*Likely outcome*<br>Biodiversity value to native fauna is predicted to be small as poplar are exotic. As poplar are deciduous, predicted to have less value to biodiversity than an evergreen tree. | Guevara-Escobar et al. [26] | Yes | *Evidence for*<br>16 native and exotic bird species documented in high density (no density was given but the canopy was stated to be dense) native forest stands of kānuka. Higher density forest stands host diverse invertebrate populations.<br>*No evidence*<br>As far as we are aware, nothing is known about how kānuka influences fungi, bird or insect populations in a silvopastoral system.<br>*Likely outcome*<br>Although only high density kānuka stands (>1000 trees ha$^{-1}$) have been studied, a kānuka silvopastoral system is predicted to have a high biodiversity value to native fauna as the genus is native. | Williams and Karl [39]; Boffa Miskell Limited [50] | Yes |
| Greenhouse gas implications | *Evidence for*<br>The above and below ground carbon pool of a poplar silvopastoral system was estimated to be 18.1 tonnes ha$^{-1}$. Nevertheless, the amount of carbon sequestered (above ground biomass) would reduce after the tree is felled.<br>*Evidence against*<br>No clear evidence poplars increase soil organic matter beneath their canopies.<br>*No evidence*<br>It is unknown how a poplar silvopastoral system may influence methane and nitrous oxide emissions. | Guevara-Escobar et al. [26]; Wall [27] | Yes | *No evidence*<br>It is unknown how kānuka impacts soil conditions and the carbon pool of a kānuka silvopastoral system has not been estimated. Is unknown how a kānuka silvopastoral system may influence methane and nitrous oxide emissions.<br>*Likely outcome*<br>If kānuka can grow for > 100 years in hill country, it would be a long-term carbon sink in terms of above and below ground biomass when compared to hill country without trees. | | Yes |

**Table 3.** *Cont.*

| Silvopastoral Outcome | Poplar (*Populus* spp.) Outcome | | Priority Area for Research | Kānuka (*Kunzea* spp.) Outcome | | Priority Area for Research |
|---|---|---|---|---|---|---|
| Soil conservation effectiveness | *Evidence for*<br>Highly effective as soil conservation trees due to their large total root length, lateral root spread (even when not fully-grown), as well as their high root tensile strength. One study found poplar to have an average maximum effective distance of 20 m for landslide mitigation. | Hawley and Dymond [65]; Douglas et al. [66]; McIvor [12]; Spiekermann et al. [13] | No—the soil conservation effectiveness of poplar is well understood. | *Evidence for*<br>Even though root systems of 20–200 trees ha$^{-1}$ have not been studied, one study found kānuka to have an average maximum effective distance of 17.0 m for landslide mitigation.<br>More research is required on the root distribution of kānuka growing at low densities (20–200 tree ha$^{-1}$) to gain a better understanding of the soil conservation value of a kānuka silvopastoral system. | Spiekermann et al. [13] | Yes |
| Time until the provision of a silvopastoral outcome | *Evidence for*<br>Quick as poplar are fast growing. | McIvor et al. [42] | No | *No evidence*<br>There is no quantitative information on the growth rate of kānuka or kānuka roots growing at low densities (20–200 trees ha$^{-1}$).<br>*Likely outcome*<br>Slower than poplar, as poplar are a fast-growing tree, and one qualitative study provides evidence that kānuka grows more slowly than poplar. | Boffa Miskell Limited [50] | Yes |

**Table 3.** *Cont.*

| Silvopastoral Outcome | Poplar (*Populus* spp.) Outcome | Priority Area for Research | Kānuka (*Kunzea* spp.) Outcome | | Priority Area for Research |
|---|---|---|---|---|---|
| The ability of farmers to receive extra income streams from tree plantings | *Evidence for*<br>Fodder—feeding poplar fodder to livestock is a practice undertaken by some farmers in summer drought conditions.<br>Emissions trading scheme—poplars at 30% canopy are eligible for carbon credits.<br>*Evidence against*<br>Wood—although poplars can be pruned and harvested for timber, as of 2021, this isn't a regular practice in New Zealand. | Charlton et al. [25]; Kemp et al. [2] | No | *Evidence for*<br>Emissions trading scheme—kānuka at 30% canopy are eligible for carbon credits.<br>*No evidence*<br>Timber—the commercial value of kānuka wood (for firewood and timber) is unknown. It is suggested that harvesting kānuka for timber is not a suitable practice for a kānuka hill country silvopastoral system because the tree density will be low (< 200 trees ha$^{-1}$) compared to a typical plantation density, plus when the trees are felled this would stop each tree's impact on other silvopastoral outcomes.<br>Honey—high density stands of trees > 40 ha are generally required to harvest high purity kānuka honey so it is unknown if honey can be harvested from a low density (20–200 trees ha$^{-1}$) kānuka silvopastoral system. Further research is required.<br>Essential oil—it is unlikely that a kānuka silvopastoral system would provide enough foliage for essential oil production because of the low density (20–200 trees ha$^{-1}$), although further research is required to confirm this. | Boffa Miskell Limited [50] | Yes—more information on the commercial potential of kānuka wood, honey and essential oil production is required. |

**Table 3.** *Cont.*

| Silvopastoral Outcome | Poplar (*Populus* spp.) Outcome | | Priority Area for Research | Kānuka (*Kunzea* spp.) Outcome | | Priority Area for Research |
|---|---|---|---|---|---|---|
| Ease of pruning, management and harvesting tree products | *Evidence against* Tall height and multi-branching habit mean management is difficult and often dangerous. | Charlton et al. [25] | No—there are other outcomes which have a higher priority. | *No evidence* *Likely outcome* The smaller and compact habit of kānuka compared to poplar suggests management would be easier. | | No—there are other outcomes which have a higher priority. |
| Cultural values | *No evidence* As far as we are aware, there has been no research on the cultural value of poplar, despite there being a lot of research on the functional value of poplar. *Likely outcome* Poplar is an exotic genus so it is predicted to have less value than a native genus. | | Yes | *Evidence for* Kānuka is a native and so has cultural significance. Nevertheless, more work is required to understand the cultural significance of kānuka compared to other genera (native or exotic) in New Zealand. | | Yes |
| Aesthetics | *Evidence against* One study has shown that when people are informed that shelterbelts are exotic, they are preferred less than native shelterbelts. *No evidence* As far as we are aware, there have been no studies on how the preference of poplar compares to other genera. | Brown et al. [67] | No—despite little research, there are more important research priorities for poplar. | *Evidence for* One study has shown that when people are informed that shelterbelts contain native trees, they are preferred over exotic shelterbelts. *No evidence* As far as we are aware, there have been no studies on the visual qualities of specific trees within a native tree category, or on kānuka specifically. | Brown et al. [67] | No—despite little research, there are more important research priorities for kānuka. |

**Table 3.** *Cont.*

| Silvopastoral Outcome | Poplar (*Populus* spp.) Outcome | | Priority Area for Research | Kānuka (*Kunzea* spp.) Outcome | | Priority Area for Research |
|---|---|---|---|---|---|---|
| Longevity of the trees | *Evidence against*<br>Tall height and multi-branching habit mean they are not very resistant against wind damage<br>Best management practice suggests felling and replanting trees after 40 years (due to impact of wind on branches, and wood rot or leaf fungus).<br>Above ground silvopastoral benefits are lost when the trees are felled.<br>It is unknown how resistant new straighter cultivars are against wind as they have only recently been planted. | Charlton et al. [25] | Yes—an understanding of the resistance of new cultivars to wind damage is important. | *No evidence*<br>*Likely outcome*<br>The small and compact habit of kānuka compared to poplar, that they are native to windy hill country conditions, and are already found on many parts of hill country, suggests kānuka are highly resistant against wind damage.<br>If kānuka can grow up to 400 years in hill country, even if only over 100 years, this means silvopastoral benefits will be lasting compared to poplar. | Boffa Miskell Limited [50] | Yes—confirming the longevity of kānuka is important. |
| Costs and ease of establishment | *Evidence for*<br>Planting as unrooted poles is an efficient way of planting trees. Recommended practice is excluding large cattle for 2 years, but sheep can still be grazed. Survival rate is normally high for poplar.<br>Costs $20–25 NZD to plant a pole as of 2021 (not including labour and transport costs).<br>*Evidence against*<br>The survival of poplar can be low, and more detailed quantitative information is required to understand the instances when survival rates can be low.<br>*No evidence*<br>More work is required to understand the establishment of poplar on the steepest hill country slopes. | Marden and Phillips [49]; McIvor [59] | Yes—more research is required on the establishment of poplar on steeper, more erosion prone slopes. | *Evidence against*<br>The time required to plant seedlings and protect them is longer than when planting poplar poles.<br>Cost of planting and protecting a commercially bought 50 cm kānuka seedling with protection is $20–30 NZD as of 2021 (not including labour and transport costs).<br>*No evidence*<br>Nevertheless, there is limited understanding into the methods of establishing kānuka in hill country, and more work is required to better understand kānuka establishment. | | Yes—comparing the establishment ease of kānuka with poplar is a priority as it is an important outcome in hill country. |

**Table 3.** *Cont.*

| Silvopastoral Outcome | Poplar (*Populus* spp.) Outcome | | Priority Area for Research | Kānuka (*Kunzea* spp.) Outcome | Priority Area for Research |
|---|---|---|---|---|---|
| Special qualities reducing animal interactions with the tree (thorniness, bitter leaves, etc.) | *Evidence for* Poplar can be established as unrooted poles which reduces the chance of grazing by livestock, as when leaves grow on the poles, they are normally above the reach of grazing livestock. | Marden and Phillips [49] | No | *No evidence* *Likely outcome* If kānuka are browsed less than other genera due to their leaves being bitter, establishing the seedlings or young trees may require protection for a shorter period of time than other more desirable browse genera. | Yes—understanding the interaction between kānuka and livestock will be useful information when attempted to establish kānuka. |
| Ability to refine the tree form for improved silvopastoral outcomes | *No evidence* *Likely outcome* Even though pruning, coppicing, and pollarding is possible that will reduce management in later life, this is only done sparingly by farms. | | No—there are other outcomes which have a higher priority. | *No evidence* It is unknown how a refined form will impact hill county silvopastoral outcomes, or if tree management would be taken up by landowners. | No—there are other outcomes which have a higher priority. |

## 4. Key Comparisons between Poplar and Kānuka

The following section explains in more detail the key comparative findings from the framework for important poplar and kānuka silvopastoral system outcomes.

### 4.1. The Interaction of Poplar and Kānuka with the Pasture and Soil

A disadvantage of poplar is the reduced pasture growth beneath their canopies [10]. There is no clear evidence whether poplar positively influence the water or nutrient dynamics of the agricultural system [26,27,63,64]. Possible attributes responsible for this competitive relationship with pasture could be their high-water use [51], their large and spreading form discouraging preferential grazing beneath their canopies, or their large canopy causing too much shading, in addition to their deciduous nature causing grass smothering [2,62], potentially reducing animal nutrient transfer in winter or reducing their influence on winter temperatures beneath their canopies [64]. Some of these factors are explored below.

Water use of fully-grown individually spaced poplar trees in hill country (37.2 stems ha$^{-1}$) was investigated by Guevara-Escobar et al. [51]. They found that average individual tree water use was 180.1 L day$^{-1}$ during a spring study period, which equates to an equivalent water use of 1.2 mm day$^{-1}$. The maximum water use over their tree repetitions was 417.0 L day$^{-1}$. A review of tree water use for 67 species (including hybrids) by Wullschleger et al. [68] suggests that the average water use by poplars in the Guevara-Escobar et al. [51] study of 180.0 L day$^{-1}$ is high. As well as having high water use, 6-month-old *Populus euramericana* (Guinier) trees were shown to have isohydric behaviour in which leaf water potential was maintained in well-watered, medium deficit, and severe deficit soil conditions [69]. Therefore, if the poplar cultivars planted in New Zealand also show isohydric behaviour, even in severe deficit soil conditions, they will have the same high water use as in saturated soil conditions [69]. Poplars cannot be compared with kānuka in terms of water use as the water use of kānuka is unknown. Nevertheless, if kānuka does use less water than poplar, this may be beneficial in terms of reducing tree-pasture water competition in the silvopastoral system.

Poplars are deciduous and lose their leaves in autumn, reducing the ability of the tree canopy to buffer air temperatures during winter months, influencing pasture growth and animal shelter effects. This was confirmed by Guevara-Escobar et al. [64] who did not find evidence that poplar buffer winter temperatures. Kānuka trees, however, are evergreen, and maintain their foliage year-round. Previous research presents examples of trees in agroforestry systems buffering winter air temperatures beneath their canopies. In Central-Western Spain, Moreno et al. [70] found the daily minimum air temperature to increase from 7.4 °C 1 m from the trunk to 6.3 °C 20 m from the trunk in the *dehesa* system of Spain. In a *Paulownia* spp. silvo-arable system in Eastern-central China, mean winter air temperature was 0.2–1.0 °C higher under trees compared with open cropping land [71]. Based on these findings, we postulate that having an evergreen tree canopy over hill country pastures in winter could buffer winter air temperatures, and this may result in increased pasture growth when temperature may otherwise limit growth.

The presence of tree canopies during winter is likely to attract more animals as they seek shelter from colder temperatures and wind. As animals are a key mechanism for nutrient transfer in hill country [72], if animals do preferentially spend time beneath kānuka in winter, this could have important implications for nutrient build-up beneath the canopy. Moreover, this should have animal health benefits, in addition to potentially reducing live weight losses as less energy may be used maintaining body temperatures. On the contrary, a canopy during winter will reduce the amount of light reaching the ground or pasture. If light limits growth during winter months, this could negatively affect pasture growth. Additionally, if animals spend too much time beneath winter canopies under certain trees, this may result in excess animal camping, potentially resulting in soil compaction and grass smothering.

In agroforestry systems, trees add leaves to the soil which can help build up soil organic matter (SOM) and nitrogen [73,74]. For *Populus maximowiczii* Henry × *P. nigra*

L. cultivars, Douglas et al. [62] recorded 1.7 t DM ha$^{-1}$ year$^{-1}$ of leaf fall in unevenly planted poplar stands of 25–400 stems ha$^{-1}$. Douglas et al. [62] found open pasture to have more annual grass biomass (8.5 t DM ha$^{-1}$) than grass plus poplar leaf biomass under poplar (8.3 t DM ha$^{-1}$). The reduction in pasture growth beneath the canopies (6.6 t DM ha$^{-1}$ year$^{-1}$) was not compensated by the 1.7 t DM ha$^{-1}$ year$^{-1}$ addition of poplar leaves. Moreover, the nutritional value of recently shed poplar leaves have not been studied and although their fodder quality is good [25], after the leaves have begun to decompose they would most likely have a lower nutritional value when compared to green leaves in the canopy. Kānuka leaf fall has been measured in high-density unmanaged mixed stands of kānuka and mānuka (no density was given, see Figure 3) [75]. The two sites in the study had an average leaf fall of 2.2 t DM ha$^{-1}$ year$^{-1}$. In contrast to poplars, this leaf fall occurred throughout the year, which may potentially reduce grass smothering during autumn, and as fewer leaves may be grazed by animals, increase the amount of organic matter that is recycled back into the soil.

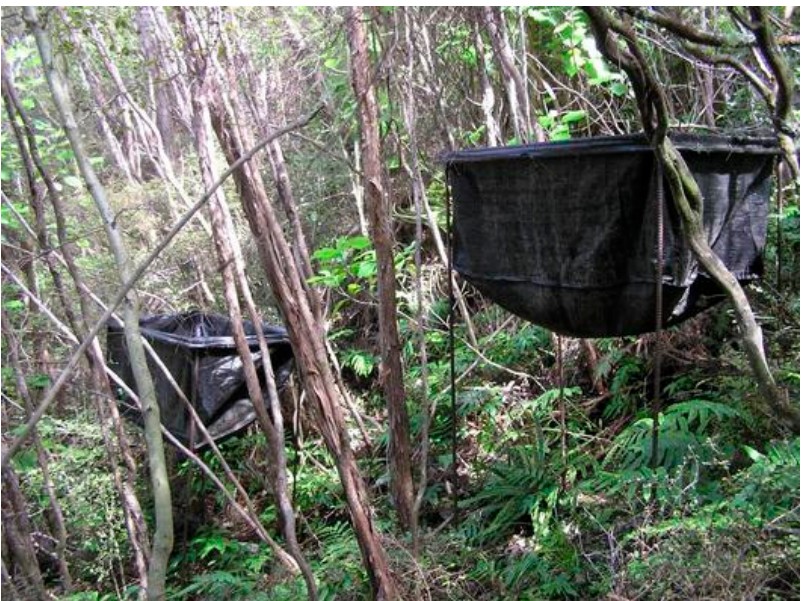

**Figure 3.** A high-density mānuka-kānuka shrubland study site for the leaf fall study by Lambie and Dando [75] (pp. 612).

These factors suggest kānuka could have a facilitating relationship, as opposed to poplar's competitive relationship, with the hill country pastoral environment. If kānuka is found to positively influence soil conditions beneath their canopies, this may have important implications for the productivity of low producing hill country, as well as for soil erosion and the hydrology of the system. Comparing the agricultural environment beneath poplar with genera of contrasting attributes would be informative and help disseminate which attributes may be leading to the negative interaction between poplar and pasture.

*4.2. Longevity*

The 25 m or greater height and spreading branches of older poplar cultivars (high vulnerability to branch windbreak), and susceptibility to leaf rust and fungus, mean their longevity is low (~40 years) [25] when compared to holm oak (*Quercus ilex* L.) trees of the *dehesa* agroforestry system in Southern Europe (trees can grow for 100 to 250 years) [76]. This impacts the long-term influence of each tree and requires that trees are felled and replanted. This represents a cost to the farmer or taxpayer and likely reduces the long-term carbon sequestration potential of each tree. The framework presents evidence that kānuka may have a similar longevity to holm oak as (1) they can potentially grow up to 400 years old [50], (2) hill country is their ecological range, meaning that they may be less susceptible

to disease (further research is required to understand the threat from myrtle rust) and (3) their relatively shorter height will most likely result in less branch windbreak. The form and longevity of new poplar cultivars that have been developed to grow in a straighter form is unknown.

### 4.3. Establishment

The ability to plant poplar as 2–3 m unrooted poles is a major advantage of the system, as sheep and small cattle can be grazed immediately after establishment and planting is quick in comparison to the planting and protection of commercially bought 50 cm kānuka seedlings. This is a major advantage of poplar, although if kānuka was adopted as a silvopastoral tree, planting technology would most likely improve with increased demand. The cost of planting and protecting 50 cm kānuka seedlings is comparable to poplar pole establishment (not including labour and transport costs).

### 4.4. Time until an Influence on the Agricultural Environment and Soil Conservation

Another advantage of establishing poplar poles is that they are already 1.5–2.5 m high when they are planted. This, in addition to their quick growth rate [49,59,77], means the time until they have an influence on the silvopastoral environment (above and below ground) is quick when compared to establishing kānuka seedlings. This quick growth rate in conjunction with their expansive root system means poplar are highly effective soil conservation trees [12,65,66]. Despite no research on the root systems of kānuka trees growing at a spacings of 20–200 trees ha$^{-1}$, tree influence modelling has shown kānuka to have an average maximum effective distance of 17 m, compared to 20 m for poplar [13]. This provides some evidence kānuka may be an effective soil conservation tree, as is the case for poplar.

### 4.5. Cultural Values

Kānuka is a native to New Zealand and as such has cultural significance. Kānuka presents an opportunity as a hill country silvopastoral tree that can potentially provide many beneficial utilisation outcomes alongside its cultural significance.

### 4.6. Bird Biodiversity

To the best of our knowledge, bird populations within poplar silvopastures have not been studied in New Zealand. This is also true of kānuka in pastured hill country, despite research showing the benefits to biodiversity of other global silvopastoral systems [3,78].

Agricultural landscapes in New Zealand hold great potential to harbour high diversity [79–82]. Blackwell et al. [79] found 'conventional' sheep and beef farms had significantly greater species abundance (total number of birds recorded) and diversity (number of different species) than all other studied landscape types—native forest, scrub, pine plantations, and kiwi-fruit orchards. However, there were two to three times fewer native species on the sheep and beef farms and kiwi-fruit orchards compared with native forest, pine plantation, and scrub. A similar conclusion was found for arable land in the South Island: species diversity was similar for native (16) and introduced (17) birds over a 2-year study, although species richness was much higher for introduced bird species (winter: $9.3 \pm 0.4$; breeding season: $11.2 \pm 0.4$) compared to native species (winter: $3.3 \pm 0.2$; breeding season: $1.7 \pm 0.2$) [80].

Although native bird populations have been shown to be smaller in productive landscapes, Blackwell et al. [79] found bird richness variation within sheep and beef farms. The number of native birds increased on farms which had more woody vegetation. Blackwell et al. [79] (pp. 70) conclude that there is great potential for "production landscapes to be flush with biodiversity" if there is more woody vegetation growing on New Zealand productive landscapes.

Williams and Karl [39] reported that a dense canopy of kānuka supported 15 bird species, with korimako/bellbird (*Anthornis melanura*), pīpipi/brown creepers (*Mohoua*

*novaeseelandiae*), and riroriro/grey warbler (*Gerygone igata*) being most common in the kānuka stands compared to gorse stands. This study gives some evidence that the native and evergreen nature of kānuka may present an opportunity for enhancing the connectivity of New Zealand's forested ecosystems within agricultural landscapes.

*4.7. Additional Income*

Honey from mānuka has been medically popularised because of its non-peroxide anti-bacterial properties [83,84]. Kānuka also has anti-bacterial properties and can be used as an antiseptic on wounds [55]. In addition, kānuka honey has been shown to have anti-viral [53], immunostimulatory [54], and anti-inflammatory properties [56].

To produce un-diluted honey which maximises these beneficial properties, bees must harvest as much nectar as possible from the flower of interest. Mānuka is currently commercially produced at a large scale in New Zealand, requiring monocultures of mānuka greater than 40 ha to achieve desired honey quality [50]. If kānuka was growing singly-spaced within a pasture system, there is a high chance other flowers would be available to foraging bees, such as clover (*Trifolium* spp.) or gorse, diluting the quality of honey produced. Boffa Miskell Limited [50] (pp. 19) does state that some interviewed farmers suggested grazing adjacent pastures very low during the flowering season to reduce nectar dilution, although "there are no data to verify the effectiveness of this strategy".

Essential oil can be produced from kānuka leaves. Recent research has explored the use of this essential oil as an eco-friendly pesticide for aphid populations [57] and *Drosophila suzukii* [58] with encouraging results. Leaves and branches under 10 mm in diameter are harvested every 3 to 5 years from trees up to 7 years old, as these trees have the greatest leaf:shoot ratio and the tree height ensures ease of harvest [85]. It is unlikely that a silvopastoral system with a 20 to 200 trees ha$^{-1}$ density of kānuka would provide enough foliage for economic essential oil production.

Landowners can earn credits for sequestering carbon (1 NZU = 1 tonne of sequestered $CO_2$) from the atmosphere through planting trees [52]. These NZUs can be traded based on a market-driven unit price. Although research is required to confirm this, it is likely that kānuka and poplar planted at 20 to 200 trees ha$^{-1}$ would cover the 30% land area threshold required for farmers to be able to receive NZUs. As the NZU price increased by over 1000% from 2013 to 2020 [86,87], this could become a valuable revenue opportunity for farmers who wish to maintain their land in pastoral farming.

## 5. Evaluation of the Framework

A major benefit of this new framework is that it considers visible and known tree attributes so the potential benefits and costs of a particular genera can be assessed before research is undertaken. This is important in the case of kānuka, as it has received little research in a silvopastoral context to date, even though the framework provides evidence that kānuka has many benefits in certain outcomes when compared to poplar. This provides the means to 'screen' genera quickly before undertaking resource-intensive research. Moreover, it clearly highlights the tree attribute differences which may be causing alternate silvopastoral outcomes. Trees can then be more rigorously compared and selected based on these attributes.

As tree genera differ in their attributes and outcomes, it is apparent each genus will have distinct advantages and disadvantages. Viewing silvopastoral trees as a set of attributes and subsequent outcomes clearly shows them as 'a set of *trade-offs* rather than a real *solution*' [88] (pp. 14, emphasis in original). Kānuka will not be a panacea species, nor will any other. Nevertheless, by presenting species using this novel framework, with their advantages and disadvantages clearly conveyed, this will result in more informed silvopastoral research directions.

Species and cultivars within genera will also have different attributes, as outlined in a poplar and willow planting guide [25]. This will most likely be the case for the seven

viable species of kānuka. Nevertheless, this was a level of detail beyond the scope of this review, although the framework could be used for within-genus comparisons.

A given tree's outcome may vary in differing growing situations, such as between pastoral livestock types, climates, topography or within-farm environments. The framework could also be used to compare the same species or genera in these differing environments.

One limitation of the framework being used as it is in this paper is the limited space within the tables. Using a table format does not present itself well to a more descriptive comparison between species for outcomes where little information is known. This was rectified in this paper by having a more descriptive comparison section below the tables. One solution to this would be to put the framework into a database, which could clearly show evidence for and against an outcome and provide an opportunity for more descriptive information in a notes section of the database.

Additionally, using the framework in a table format would be difficult when more than two species or genera were assessed, as the tables would most likely become cluttered with the information. Using the framework in a database would be very beneficial if more than two species or genera are compared.

When comparing the two genera, it would be helpful to add a common unit account for each of the outcomes so they can be quantitatively compared. Nevertheless, as stated in the introduction, this was beyond the scope of this review as the use of the framework in this paper is to review poplar and kānuka as silvopastoral trees and inform future research directions, not to provide a tool for quantitively evaluating tree planting decisions. This would be a valuable use of the framework, however, if enough information was available for specific species or genera.

Some of the outcomes could have had sub-categories, especially for 'above ground tree habit and phenology'. We decided not to use sub-categories because only this first outcome really warrants them, and we felt it was important to keep all the outcomes consistent. Moreover, the potential sub-categories such as 'impact of the silvopastoral tree on pasture', 'impact of the silvopastoral tree on the soil', and 'impact of the silvopastoral tree on water' are very much interlinked, as the soil is related to the availability of water, and both the soil properties and the availability of water are related to pasture growth. By maintaining one larger group, this allows for a summary statement at the end of each category, and makes it clear the holistic nature of this outcome.

Finally, as is the case in systems, many of the outcomes themselves interact with each other. For instance, 'the time until the provision of a silvopastoral outcome' interacts with 'the influence of the tree on the pasture and soil', 'livestock shelter', and 'soil conservation effectiveness', and 'the influence of the tree on the pasture and soil' interacts with 'water sediment and nutrient gains and losses'. Nevertheless, outcome-outcome interactions were not included in the framework as the focus of the framework is how the tree attributes relate to silvopastoral outcomes. We think research prioritisation and tree selection for researchers and land managers will be guided specifically by the presentation of the outcomes and their interactions with tree attributes.

## 6. Conclusions

Silvopastoralism is a land management tool that can offer holistic solutions to degraded agricultural landscapes. For silvopastoral systems to be researched, assessed, and compared in a holistic manner, a framework that outlines all their known silvopastoral outcomes is required. Moreover, by relating bio-physical tree attributes to these silvopastoral outcomes, tree selection for research and planting can be optimised based on a system's outcome needs.

The framework gives emphasis to the plethora of beneficial influences of trees to silvopastoral systems that are often not considered by New Zealand land managers, such as shelter provision, longevity, extra income from trees, the benefits of a winter tree canopy, the system's hydrology, and habitats for local fauna populations. This process clearly conveys the complexity of silvopastoral systems and extends the focus beyond more

commonly researched outcomes (pasture production and soil conservation in the case of New Zealand hill country).

The framework was then used to review specific silvopastoral systems, highlighting research gaps and generating research priorities. In a New Zealand hill country case study, this paper shows the potential value of kānuka as a silvopastoral genus, a tree with a very different set of tree attributes to poplar, the most commonly planted silvopastoral tree in hill country. Kānuka may have improved outcomes in terms of pasture production, longevity, biodiversity value, shelter and ease of management due to its smaller size, evergreen nature and that it is native to hill country. Nevertheless, more research is required on kānuka to better understand these benefits and inform its use.

There also remain many outcome knowledge gaps for poplar used in a 20–200 trees ha$^{-1}$ silvopastoral system such as biodiversity interactions, livestock shelter, greenhouse gas implications and water and nutrient gains or losses. This is surprising due to the amount of research that has been done on poplar and its widespread use. If poplar is to be fully evaluated and more fairly compared with other genera, researching these other silvopastoral outcomes is essential.

**Author Contributions:** Conceptualization, T.H.M.-S.; writing—original draft preparation, T.H.M.-S.; writing—review and editing, L.B., J.R., I.F.L. and C.P.; supervision, L.B., J.R. and I.F.L. All authors have read and agreed to the published version of the manuscript.

**Funding:** This research was funded by the doctoral scholarship programme from Massey University, New Zealand.

**Institutional Review Board Statement:** Not applicable.

**Informed Consent Statement:** Not applicable.

**Acknowledgments:** Thank you to John Wiley & Sons, Incs. for permission to reproduce Table 1, to Suzanne M. Lambie for giving us permission to use the photo in Figure 3, and to the reviewers for their valuable feedback on the manuscript.

**Conflicts of Interest:** The authors declare no conflict of interest.

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
