# Peer review of "A Framework for Reviewing Silvopastoralism: A New Zealand Hill Country Case Study"

_land, doi:10.3390/land10121386_

Round 1
Reviewer 1 Report
I enjoyed reading the revised manuscript and I feel you have satisfactorily addressed my concerns and suggestions. You have clearly documented how you built upon Wood’s framework and revised it for hill country silvopasture. I appreciate the evaluation of the framework along with limitations and suggestions for improvement. The presentation of the work as a revised framework with a specific case study now elevates the value and use of this framework in a broader context beyond New Zealand.
Author Response
This reviewer provided no revisions.
Reviewer 2 Report
I found your manuscript very interesting, well structured, and prepared. However, in some moments of the review, I had the impression that it is a hybrid between a review article and a research article. Of course, a lot of data from the literature are presented, but, maybe in some places, due to the presentation, it seems to be more of a research work.
The conclusions are certainly the conclusions of a study-experience. I think there would be a need for some changes here. Certain statements (rows 495-502) do not seem to belong to the conclusions but rather in one of the other sections of the article.
I have not yet met a section of conclusions in which to include the bibliographic reference. The conclusions belong exclusively to the authors ...
Author Response
However, in some moments of the review, I had the impression that it is a hybrid between a review article and a research article. Of course, a lot of data from the literature are presented, but, maybe in some places, due to the presentation, it seems to be more of a research work.
The conclusions are certainly the conclusions of a study-experience. I think there would be a need for some changes here. Certain statements (rows 495-502) do not seem to belong to the conclusions but rather in one of the other sections of the article.
I have not yet met a section of conclusions in which to include the bibliographic reference. The conclusions belong exclusively to the authors ...
Response 1: Thank you for this point. The conclusion has been changed so it expresses the fact that the paper has created a framework for reviewing silvopastoral trees and used the framework to review two silvopastoral trees in a New Zealand hill country case study. The conclusion now briefly summarises the framework and is followed by two paragraphs which details the review findings for poplar and kānuka.
A new section (Section 5: Evaluation of the framework) was created from subsection 4.9 after the main review discussion (Section 4: Key comparisons between poplar and kānuka) that specially addresses the evaluation of the framework. The last three paragraphs from the old conclusion were added to the beginning of this new section (Section 5) and replaced by two paragraphs that detail the review findings for poplar and kānuka. Section 4.8 has been removed and now forms part of the final 2 paragraphs in the new conclusion.
By splitting section 4 and 5, this focuses the main body of the paper on reviewing poplar and kānuka (Sections 3 and 4), which is followed by a discussion specifically on the use of the framework, as per the wishes of the other reviewer (Section 5). The conclusion then summarises the framework and the main findings that have been found from reviewing poplar and kānuka.
At two points it has been made clearer that the paper is a review by stating that the framework will be used to review and compare the genera in addition to generating research priorities (lines 21 and 154). We think that these statements in the abstract and section 3, along with explicitly stating that the framework will be used to review the genera in lines 85 – 90 in the introduction and having a clearer conclusion and two separate sections for the review part (section 3 and 4), makes it clear it is a review paper.
This manuscript is a resubmission of an earlier submission. The following is a list of the peer review reports and author responses from that submission.
Round 1
Reviewer 1 Report
This manuscript assesses the research on species attribute-outcome relationships in silvopastoralism systems in hill country New Zealand, specifically with respect to kānuka and poplar trees. The authors draw on and build upon an existing framework to do this.
Without knowing the extent to which this new framework is different from the original, it is difficult to assess the contribution of the paper to the authors’ stated goal of providing an internationally novel and valuable tool for the assessment of silvopastoral trees. The authors provide a strong argument for further research on the potential role of kānuka in silvopastoralism, but the value of the framework itself is less clear.
Major issues to consider:
- It would be valuable to see the Wood framework or an explanation of which elements are novel to this framework and how they depart from the original. Each addition (or omission) should be justified based on some criteria beyond the authors’ own judgement. For example, if whole outcome categories were added, what led to this insight? Literature? Another framework?
- The above will likely help to clarify another issue, which is why a new framework is needed. If the present framework also needs to be modified for other contexts, it is not clear how this particular modification of the original framework is an important step/ development. Do the authors see it as generalizable more readily to all silvopastoral contexts?
- As written, it is somewhat unclear who the intended audience is. If the audience is primarily academic, and the paper contributing insights to support a research agenda, this is justified by the current mss but should be clarified. The paper also makes mention of a potential audience of silvopastoralists or decision-makers. The value to this population is much less clear. The framework, as presented, bears little similarity to decision tools and is not structured for straightforward comparisons between options (e.g., no units on outcomes, broad/ vague attribute and outcome categories). I can imagine a framework or decision tool that accomplishes this better, but I think that would be a very different paper. I suggest strictly focusing on the proximal value to the research community with the idea that better research may someday lead to better (or different) management decisions.
Minor issues:
- Some grammatical errors throughout—see attached pdf.
- The framework might feel more useful if the attributes were free to interact with multiple outcomes, as the authors mention is indeed the case. Instead of a table, this could look like arrows of varying widths, or something similar, see e.g. page iv (https://www.millenniumassessment.org/documents/document.356.aspx.pdf)
- Some of the attribute categories seem either a poor fit for what is included or too broad to be useful, e.g. “site adaptability and ecological range” would perhaps better be “site suitability” related to “pests and vulnerabilities” and “longevity/ mortality rates”. “Special qualities” could have “ecological symbioses (incl. nitrogen fixation), herbivory defense, etc”
- To be more helpful to a research agenda, I would recommend adding subcategories to outcomes, too. E.g. Pasture, animals, and soil could be: “beneath canopy: soil quality, soil moisture” etc, focusing on mechanism connecting attributes with outcomes, to increase usefulness (e.g. how do poplars reduce pasture?) This is done well with the subsequent sub-categories
- Column in framework reading” research required” seems strange; maybe change to "Priority area for research"? Some of the evidence has very few cases to support it, indicating that more research would be helpful, but perhaps not a priority.
- Columns of research could be changed to give more information about certainty, e.g. “evidence for, evidence against, certainty, missing contexts” or similar.
- The last comment in Poplar “site adaptability and ecological range” seems like it belongs more/ equally in the following category of establishment method….
- “Cultural values” on page 10 are not justified and certainly need to be, especially if the authors plan to assert that no research is required to address this. There is a significant pool of research on cultural ecosystem services and their valuation and the links between exotic vs. native species and their cultural value is far from being established in that literature. Citations required. Same goes for “aesthetics” on page 11.

Reviewer 2 Report
Overall Comments
The manuscript describes a well-constructed process for evaluating species for silvopastoral uses that should prove useful for producers and land managers in New Zealand. The manuscript also provides a good summary of research gaps on the use poplar and kānuka for silvopastoral systems in New Zealand. The framework could have value for other silvopastoral systems beyond New Zealand.
That said, I think the paper could be stronger and have more international utility by highlighting the framework and using the species comparison more as a case study. Since this manuscript is being submitted to a special issue on Mountains Under Pressure, the framework can be pitched as tool for supporting appropriate species selection in mountainous regions with a focus on silvopastoral systems. Currently, the emphasis is more on the species and less on the framework. Consequently, the manuscript tends to come across as a review article on two species, one endemic to New Zealand which may have limited interest and value in an international journal.
I would consider restructuring the introduction to better emphasize the need for an evidence-based framework for plant selection for silvopastoralism in hill country or mountainous regions. This would fit better with the purpose of the special issue and would help clarify in the beginning that the framework is the innovative and transferable piece of this research.
I would consider moving the section 3. A framework for hill country silvopastoralism in front of any significant discussion of plant species. The results section (section 4) then becomes application of the framework using poplar and kānuka as a case study and the discussion (section 5) is the assessment of the framework. It would be useful to have more information on the evaluation of the framework. What worked well and what didn’t? Limitations? Are there lessons learned for others to use the framework? Text in section 5 could be shorten since much of the information was already presented in the tables. This will allow for room to provide more of an evaluation of the framework.
Since the framework is the key piece of the manuscript, you might want to provide a more detail on reduction of Wood’s 15 attributes to your 9 attributes. I think it would be worthwhile to point out that Wood did not provide an actual example of his framework being applied. The value of your manuscript is building on Wood’s work, refining the framework for a specific application, and evaluating the framework with specific species using available evidence.
This would also help lessen the negative bias in the introduction towards poplar and willow. The current introduction can give readers the impression that one is already bias in comparing the species. Allow your objective and evidence-based framework to demonstrate which species can offer the most overall benefits.
Tree Attributes
Below ground tree habitat. While the rooting length and tensile strength is important for slope stabilization, is there a need to consider nutrient and water competition between the trees and forage species? Maybe this is an area that needs additional research.
May refer to:
- Sharrow SH (1999) Silvopastoralism: competition and facilitation between trees, livestock, and improved grass-clover pastures on temperate rainfed lands. In: Buck LE, Lassoie J, Fernandez ECM (eds) Agroforestry in sustainable agricultural systems. CRC Press, Boca Raton, pp 111–130
- Tree Selection Guide for Mid-Atlantic Silvopastures: https://vtechworks.lib.vt.edu/bitstream/handle/10919/87405/Beegle_DK_T_2019.pdf?sequence=1&isAllowed=y
Table 3 pg 9. Not sure the connection between bitter leaves and shorter protection period and nitrogen fixation?
Aesthetics is not necessarily related to exotic vs native. Is there any research on visual preference of New Zealand landscapes?
Other comments
Reference #34 has the chapter and book citation backwards.